# The Impact of Personal Thoracic Impedance on Electrical Cardioversion in Patients with Atrial Arrhythmias

**DOI:** 10.3390/medicina57060618

**Published:** 2021-06-13

**Authors:** Seung-Young Roh, Jinhee Ahn, Kwang-No Lee, Yong-Soo Baek, Dong-Hyeok Kim, Dae-In Lee, Jaemin Shim, Jong-Il Choi, Young-Hoon Kim

**Affiliations:** 1Division of Cardiology, Department of Internal Medicine, Korea University College of Medicine and Korea University Medical Center, Seoul 02841, Korea; rsy008@gmail.com (S.-Y.R.); jaemins@korea.ac.kr (J.S.); jongilchoi@korea.ac.kr (J.-I.C.); 2Division of Cardiology, Department of Internal Medicine, Pusan National University Hsopital, Busan 49241, Korea; reinee7@gmail.com; 3Division of Cardiology, Department of Internal Medicine, Ajou University Hospital, Suwon 16499, Korea; knlee81@gmail.com; 4Division of Cardiology, Department of Internal Medicine, Inha University hospital, Incheon 22332, Korea; existsoo@hanmail.net; 5Division of Cardiology, Department of Internal Medicine, Ewha University Hospital, Seoul 07804, Korea; tomas9912@naver.com; 6Division of Cardiology, Department of Internal Medicine, Chungbuk National University Hospital, Chungju-si 28644, Korea; acttopia@gmail.com

**Keywords:** cardioversion, heart failure, atrial fibrillation, atrial tachycardia, thoracic impedance

## Abstract

*Background and Objectives*: Direct current cardioversion (DCCV) is a safe and useful treatment for atrial tachyarrhythmias. In the past, the energy delivered in DCCV was decided upon empirically, based only on the type of tachyarrhythmia. This conventional method does not consider individual factors and may lead to unnecessary electrical damage. *Materials and Methods*: We performed DCCV in patients with atrial tachyarrhythmias. The impedance and electrical current at the moment of shock were measured. The human thoracic impedance between both defibrillator patches and the electric current that was used were measured. *Results*: A total of 683 DCCVs were performed on 466 atrial tachyarrhythmia patients. The average impedance was 64 ± 11 Ω and the average successful current was 23 ± 6 mA. The magnitude of the electrical current that was successful depended upon the human impedance (linear regression, B = −0.266, *p* < 0.001) and the left atrial diameter (B = 0.092, *p* < 0.001). Impedance was directly proportional to body mass index (BMI) (B = 1.598, *p* < 0.001) and was higher in females than in males (77 ± 15 Ω vs. 63 ± 11 Ω, *p* < 0.001). Notably, the high-impedance (>70 Ω) group had a higher BMI (27 ± 4 kg/m^2^ vs. 25 ± 3 kg/m^2^, *p* < 0.001) and a higher proportion of females (37% vs. 9%, *p* < 0.001) than the low-impedance group (<70 Ω). However, thoracic impedance was not an independent predictor for successful DCCV. *Conclusions*: Human thoracic impedance was one of the factors that impacted the level of electrical current required for successful DCCV in patients with atrial arrhythmias. In the future, it will be helpful to consider individual predictors, such as BMI and gender, to minimize electrical damage during DCCV.

## 1. Introduction

Direct current cardioversion (DCCV) is one of the most widely used methods for terminating symptomatic atrial arrhythmias, including atrial fibrillation (AF). Many studies have shown that DCCV is effective and safe since Lown et al. first introduced DCCV in 1962 [1,2,3,4,5]. However, no consensus or recommendations about the initial energy level for DCCV exist, despite DCCV’s common and practical use. Usually, the initial energy level for the shock is determined based on the type of arrhythmia and the operator’s level of experience [6]. Unfortunately, this uncertainty has led to excessive myocardial damage and complications, such as post-cardioversion arrhythmia or coronary spasms [7,8,9,10]. DCCVs performed by experienced clinicians rarely induce complications. However, complications can increase by repeated shocks or cardioversion using excessive energy from inexperienced clinicians. In a recently published paper, 887 patients who underwent cardioversion for AF in an emergency room were analyzed. In the overall population, 6%, 10%, and 1% of patients suffered electrical complications, namely, arrhythmia, shock or desaturation, and chest wall burn, respectively, as a complication of DCCVs [11].

Most of the previous studies exploring factors that might impact DCCV focused on anatomical and electrical remodeling or medications [12,13]. According to a “current-based” electrical cardioversion approach, the density of the current delivered to the atrial myocardium is a more important determinant of myocardial damage than the shock energy level [14]. In addition, the density of the current delivered to the myocardium is related to the thoracic impedance of the human body. Thoracic electrical impedance is the inverse resistance of the crossing currents between the two electrode pads. Electrical current changes with thoracic impedance under an equivalent energy level. Thoracic impedance is known to depend on several factors, including body surface area, pad size, pad location, and pad-to-skin contact. This means that thoracic impedance can be changed depending on the components and the physical distance between both pads. Therefore, the objective of the present study was to assess the clinical implications of thoracic impedance in DCCV for various atrial arrhythmias.

## 2. Methods

### 2.1. Study Population

From January 2011 to December 2012, consecutive patients who underwent DCCV for symptomatic atrial arrhythmias were included in this observational, single-center study (Figure 1). Patients under 19 years of age, pregnant women, and patients with secondary arrhythmias (arrhythmias due to acute myocardial infarction, cardiac surgery, pericarditis, myocarditis, thyrotoxicosis, or acute pulmonary disease) were excluded. The patients were divided into five groups according to their type of atrial arrhythmia as follows: atrial tachycardia (AT), atrial flutter (AFL), paroxysmal atrial fibrillation (PAF, AF that spontaneously terminated), persistent atrial fibrillation (PeAF, AF persisting beyond 7 days), and long-standing persistent atrial fibrillation (LsPeAF, AF persisting beyond 1 year).

All patients received an oral anticoagulant for at least 4 weeks before and after cardioversion.

In order to rule out structural heart disease in all patients, trans-thoracic echocardiography (TTE) was performed within the 6 months prior to DCCV. Additionally, intra-cardiac thrombi were ruled out via trans-esophageal echocardiography (TEE) within 24 h prior to DCCV in all patients. Blood samples for serologic assays were collected and optimal anticoagulation was established before DCCV was performed. The research protocol complied with the principles of the Declaration of Helsinki and was approved by the Institutional Review Board of the Korea University Anam Hospital (2017AN0163).

### 2.2. Cardioversion Protocol

After obtaining written informed consent, DCCV was performed after the patients were sedated with intravenous boluses of propofol (1 to 3 mg/kg) or midazolam (1 mg/kg). A biphasic R-wave synchronized shock was applied to the patients via square, self-adhesive skin electrodes (TZ Medical Inc., Portland, OR, USA) placed in anterior–posterior positions [15,16,17,18]. Electrical shocks were delivered using the HeartStart MRx^®^ defibrillator (Philips, Inc.). The current intensity and thoracic electrical impedance were measured after each shock. Thoracic impedance was automatically measured between both patches after delivering the energy. There was no additional work to measure the thoracic impedance.

The initial energy level was determined based on the patient’s type of atrial arrhythmia. In patients with AT, AFL, or PAF, 50 or 70 joules (J) were delivered as the initial shock. In patients with PeAF or LsPeAF, 70 or 100 joules (J) were delivered. If the shock failed to restore sinus rhythm (SR), the energy was gradually increased to 100, then 150, and up to a maximum of 200 J. If a biphasic shock of 200 J failed to terminate the arrhythmia, 5 mg/kg of intravenous amiodarone was infused for 30 min. After the procedure, the patient was monitored using electrocardiography for at least 1 h. Successful DCCV was defined as a patient who was discharged with SR.

### 2.3. Statistical Analyses

Comparisons between groups were performed using *t*-tests for continuous data and the chi-square test or Fisher’s exact test for categorical data. Paired *t*-tests were used to measure serial electrical characteristic changes with repeated shocks in one case. The relationship between the current and the thoracic impedance was assessed using correlational analyses. Logistic regression was performed to determine the independent correlates of thoracic impedance. The relationships between the continuous variables were evaluated using linear regression models. Statistical significance was set to *p* ≤ 0.05. All statistical analyses were performed using SPSS 21.0 (SPSS Inc., Chicago, IL, USA).

## 3. Results

### 3.1. Baseline Characteristics

We performed 683 total DCCVs in 466 enrolled patients with atrial arrhythmias (60 ± 11 years old, 376 males). Baseline characteristics stratified according to each arrhythmia group are shown in Table 1. No differences in body surface area (BSA) or body mass index (BMI) were observed between the five groups. The number of patients with hypertension was different in each group (*p* = 0.005). The prevalences of diabetes mellitus, chronic kidney disease (glomerular filtration rate < 60 mL/min/1.73 m^2^), mitral valve disease, and left ventricular (LV) systolic dysfunction (estimated ejection fraction <40%) did not vary between groups. While the LV mass index (LVMI) and LV ejection fraction (EF) were similar in all groups, the mean left atrial anteroposterior diameters (LAD) of two PeAF (PeAF + LsPeAF) groups were significantly larger than that of the other three (AT + AFL + PAF) groups.

The electrical characteristics of these five groups are summarized in Table 2. In this study, the total success rate of DCCV for atrial arrhythmias was 93.3%. The SR restoration rate in the two PeAF groups (PeAF + LsPeAF) was lower than in the AT, AFL, and PAF groups. Every person had their own individual thoracic impedance. In this study, the patients’ thoracic impendence did not vary as the delivered energy level changed. The mean thoracic impedance of all patients was calculated to be 66.1 ± 12 Ω, and it was lower in the PeAF and LsPeAF groups compared to the other groups. The minimum initial thoracic impedance was 31.3 Ω, and the maximum impedance was 119.3 Ω. When DCCV successfully restored SR, the energy and the current that were delivered were found to be significantly different between all the groups (*p* < 0.001). In addition, the rate of amiodarone administration for arrhythmias that were refractory to DCCV also differed between all groups.

### 3.2. Successful Electrical Current and Thoracic Impedance

In Figure 2, the correlation between the therapeutic dose of electrical current and the thoracic impedance at the time of successful SR restoration is shown. In all patients, the electrical current delivered was inversely proportional to the thoracic impedance, regardless of the energy level that was delivered (correlation analysis, *r* = −0.593, *p* < 0.01). This correlation between the electrical current and thoracic impedance was similar in each arrhythmia-type group (AT group *r* = −0.53, AFL group *r* = −0.621, PAF group *r* = 0.667, PeAF group *r* = −0.549, LsPeAF group *r* = −0.674, *p* of all groups <0.01). The changes in impedance with repeated shocks (energy delivery) were measured in the patients who underwent more than three shocks (137 cases out of 683 total cases, 20%). The thoracic impedance gradually decreased as the shocks were repeated (paired *t*-test, Figure 3).

### 3.3. Predictor of High Thoracic Impedance

Among the differences in arrhythmia types, ages of the patients, sex, BMI, BSA, LAD, LV EF, LVMI, and amiodarone use, the multivariate linear regression analysis showed that sex and BMI were the only factors associated with increased thoracic impedance. Notably, thoracic impedance was significantly higher in female patients than in male patients (77 ± 15 Ω vs. 63 ± 11 Ω, *p* = 0.01). Impedance was also increased in patients with higher BMIs (linear regression, B = 1.558, *p* = 0.01) (Figure 4). Interestingly, thoracic impedance was not affected by arrhythmia type, age, BSA, LAD, LVMI, or amiodarone use.

Among the successful DCCV cases (*n* = 637, 93%), we also compared cases with high impedance (initial impedance ≥ 70 Ω) to those with low impedance (initial impedance < 70 Ω). The high impedance group had 421 cases (66%) and the low impedance group consisted of 216 cases (34%) (Table 3). The female-to-male ratio was significantly higher in the high-impedance group than in the low-impedance group (36.9% vs. 9.0%, *p* < 0.001). The high-impedance group showed a higher average BMI (27 ± 4 kg/m^2^ vs. 25 ± 3 kg/m^2^, *p* < 0.001) and a larger average LA diameter (46 ± 12 mm vs. 44 ± 6 mm) than the low-impedance group. The prevalence of hypertension (37% vs. 30%, *p* = 0.036) and coronary artery disease (6% vs. 3%, *p* = 0.01) was also higher in the high-impedance group compared to the low-impedance group. Furthermore, the total energy delivered and the total number of shocks tended to be higher in the high-impedance group, but these results were not statistically significant. We found that the thoracic impedance at the time of the initial shock was not significantly different between the successful DCCV and failed DCCV subgroups. Ventricular fibrillation during DCCV occurred in 20 cases (2.9%) among the entire study population, and all patients were successfully treated using immediate defibrillation. CVA occurred to two patients (0.4%) within 48 h after DCCV. Of these two patients, one patient had PAF and the other had PeAF.

## 4. Discussion

The present study found that women and patients with high BMIs had higher thoracic impedances. Notably, arrhythmia type, LAD, and antiarrhythmic drug (AAD) use did not affect thoracic impedance. Although thoracic impedance at the time of the initial shock was significantly different between all groups, multivariate linear regression analysis showed that the arrhythmia type was not associated with thoracic impedance. Furthermore, arrhythmia type had no effect on the clinical results of the DCCV.

### 4.1. The Clinical Implication of Thoracic Impedance for Successful Cardioversion

DCCV is highly successful and is safe for the treatment of atrial arrhythmias [19]. However, repeated shocks require patients to undergo longer periods of sedation and may act as a trigger for fatal arrhythmias (e.g., torsade de pointes). They may also cause malfunctioning of devices implanted in patients with intracardiac devices and dermal injuries [20,21].

The initial energy level of DCCV is conventionally determined based on the type of arrhythmia and the arrhythmia’s duration. No guidelines for determining the initial energy level exist, and therefore, the initial energy is chosen based on empirical evidence. Shocks with lower initial energy levels are sometimes useless, whereas shocks with high initial energy levels can cause myocardial and dermal injury. In this study, thoracic impedance was found to decrease with repeated shock delivery. The inflammation caused by myocardial damage is thought to be due to this decreased impedance; therefore, choosing the appropriate initial energy level is important for minimizing the number of shocks and the total energy delivered. If DCCV with the same energy is used repeatedly, the current can be increased as the impedance decreases, which can make the cardioversion successful. Restoring sinus rhythm has advantages in cardiac physiology and relieves symptoms, but may also result in shock-induced inflammation and cardiac stunning. This may lead to a temporary deterioration of heart function, even during a sinus rhythm status after DCCV.

### 4.2. The Predictor of Thoracic Impedance

It is impossible to estimate the electrical current during DCCV due to the action of normal human impedance. Electrical current and thoracic impedance showed an inverse relationship. Electrical current increases with energy level, but each person’s body has the proper thoracic impedance to counteract this increase in current. In a previous study, it was found that electrical current was related to thoracic impedance in patients with AF [22]. Our study showed that this finding applies to patients with all types of atrial arrhythmias. In addition, none of the measured comorbidities, the duration of the AF, and the AAD used were not associated with the thoracic impedance.

In this study, multivariate analysis demonstrated that a large BMI and female sex are independent predictors of high thoracic impedance. However, the mechanism by which these factors affect thoracic impedance remains unclear. Obese patients tend to have larger thoracic AP diameters and more fatty tissue, and fatty tissue has a higher impedance than lean tissue [23]. Notably, bioelectrical impedance analysis uses this difference between fatty and lean tissues to measure body component proportions [24,25]. In other words, the increased amount of fatty tissue in obese patients may have led to the observed increase in thoracic impedance. The characteristics of fatty tissue may also explain why women had higher thoracic impedances since the body fat percentage of middle-aged women is higher compared to men [26].

### 4.3. The Appropriate Level of Electrical Current for Cardioversion

Although current-based DCCV is superior to empirical, energy-based DCCV in decreasing the number of shocks and in improving safety, it is still not possible to calculate or predict the necessary therapeutic dose of electrical current for an individual patient. However, if factors that impact the thoracic impedance are found, we can predict the appropriate level of electrical current. Taking an individual’s clinical characteristics, such as thoracic fat or sex, into account may help us to determine the proper energy levels for DCCV. If cardioversion is to be performed on a female and/or a patient with a large BMI, using higher initial energy should be considered. To reduce the amount of damage to the myocardium or skin, the number of DCCVs must be reduced. Sometimes, a method that increases the contact of the heart with the pad or patch should be used. Furthermore, the ability to measure the thoracic impedance between patches in AED machines will help to determine the initial energy.

### 4.4. Study Limitations

The greatest limitation of this study is the undesignated initial shock energy level. The initial shock energy level differed according to each operator’s preferences. In most patients, just one delivered shock was enough to restore SR. Not having a designated and specific initial shock energy may act to decrease the accuracy of the data regarding energy and current at the time of success. Furthermore, we cannot explain the differences in initial thoracic impedance between the PeAF groups (PeAF + LsPeAF) and the other groups. The thoracic impedances at the time of success were different between groups due to the increase in inflammation with repeated shocks. However, the impedance at the time of initial shock was not associated with inflammation. Notably, BMI and the sex ratio did not differ between any of the groups; therefore, the impact of the differences in the initial energy levels delivered cannot be ruled out. The BMI used in this study was the median value. These results cannot be generalized because obese patients (BMI ≥ 30 kg/m^2^) were not analyzed separately.

Differences in thoracic impedance did not lead to different DCCV results. None of the clinical or electrical characteristics were significantly different between the DCCV success and failure groups and thoracic impedance did not impact the results. These results dilute the clinical significance of thoracic impedance, but the limited number of patients that were included in the failure group may have affected the ability to observe real differences in impedance.

## 5. Conclusions

Human thoracic impedance is an important factor that impacts the therapeutic dose of electrical current during the CV of all atrial arrhythmias. Considering thoracic fat or gender is helpful to customize the energy level of shocks, but not to increase the success rate of DCCV.

## Figures and Tables

**Figure 1 medicina-57-00618-f001:**
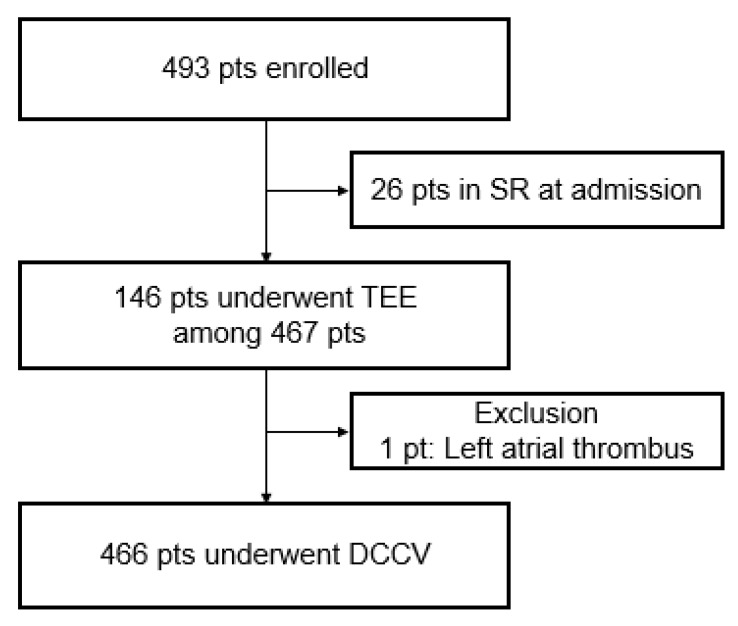
Flowchart of the study.

**Figure 2 medicina-57-00618-f002:**
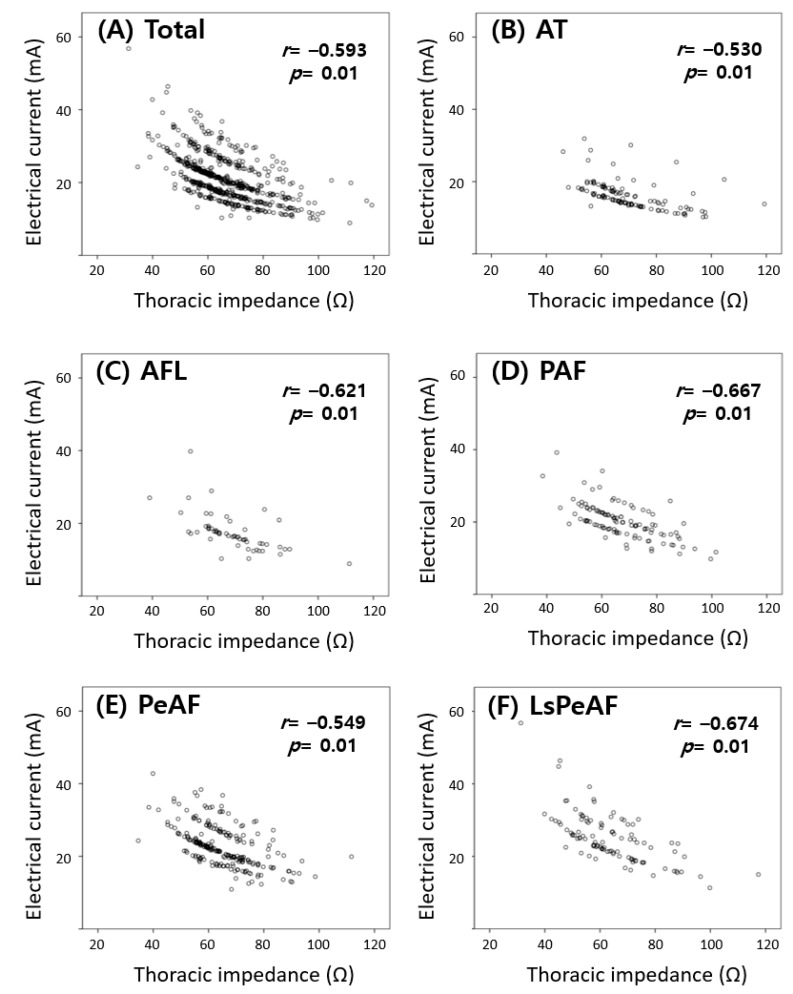
The relationship between electrical current and thoracic impedance for (**A**) all patients and patients with (**B**) atrial tachycardia, (**C**) atrial flutter, (**D**) paroxysmal atrial fibrillation, (**E**) persistent atrial fibrillation, and (**F**) long-standing persistent atrial fibrillation. As the thoracic impedance increases, the electrical current decreases.

**Figure 3 medicina-57-00618-f003:**
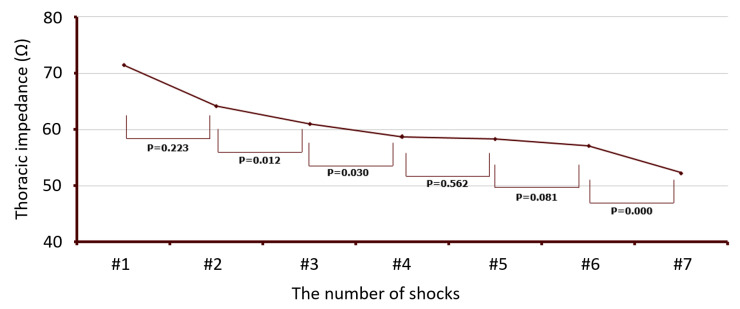
Changes in human impedance with repeated shocks in patients receiving more than three shocks.

**Figure 4 medicina-57-00618-f004:**
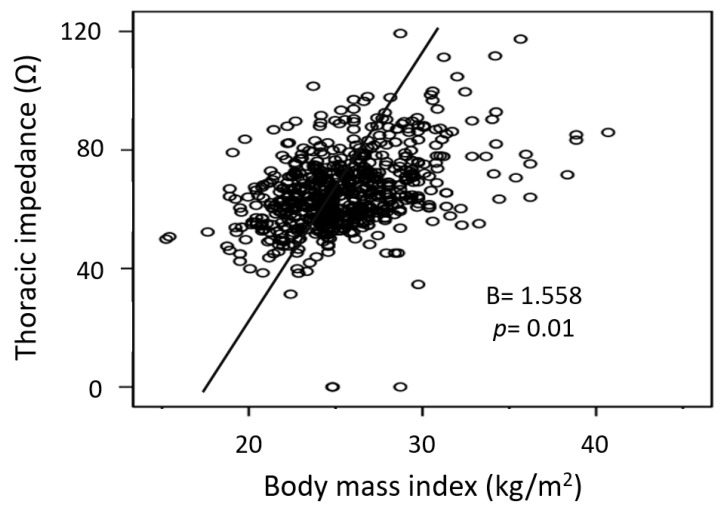
The relationship between human impedance and BMI.

**Table 1 medicina-57-00618-t001:** Baseline characteristics of the study population based on the type of arrhythmia.

	Total	AT	AFL	PAF	PeAF	LsPeAF	*p*-Value
Cases (*n*)	683	127	61	116	260	119	
Patient (*n*)	466	62	43	90	182	89	
Age (years)	60 ± 11	58 ± 11	59 ± 11	59 ± 12	59 ± 10	61 ± 10	0.302
Sex, male (%)	376 (81)	101 (80)	53 (87)	84 (72)	228 (88)	97 (82)	0.005
BMI (kg/m^2^)	25 ± 3	25 ± 3	26 ± 4	25 ± 3	26 ± 3	26 ± 3	0.52
BSA (m^2^)	1.9 ± 0.2	1.9 ± 0.3	1.8 ± 0.2	1.8 ± 0.2	1.9 ± 0.3	1.8 ± 0.2	0.56
Hypertension (%)	157 (34)	53 (42)	25 (41)	42 (36)	70 (27)	29 (24)	0.005
Diabetes mellitus (%)	41 (9)	12 (9)	6 (10)	11 (10)	16 (6)	11 (9)	0.671
Thyroid disease (%)	37 (8)	15 (12)	6 (10)	9 (8)	21 (8)	9 (8)	0.722
Chronic lung disease (%)	7 (2)	2 (2)	0	1 (1)	6 (2)	2 (2)	0.697
Valvular disease (%)	27 (6)	8 (6)	5 (8)	8 (7)	12 (5)	3 (3)	0.412
Chronic kidney disease (%)	10 (2)	1 (1)	3 (5)	2 (2)	6 (2)	2 (2)	0.443
Cerebrovascular accident (%)	28 (6)	5 (4)	2 (3)	5 (4)	12 (5)	11 (9)	0.263
Coronary artery disease (%)	14 (3)	6 (5)	3 (5)	4 (3)	8 (3)	5 (4)	0.915
Echocardiographic parameters							
LAD (mm)	45 ± 8	43 ± 11	44 ± 6	45 ± 6	45 ± 9	47 ± 6	0.004
LVMI (g/m^2^)	96 ± 25	96 ± 29	104 ± 32	96 ± 27	95 ± 23	93 ± 18	0.081
LV EF (%)		52 ± 9	52 ± 10	51 ± 9	52 ± 9	54 ± 6	0.097

Continuous variables are expressed as mean ± SD. AT—atrial tachycardia; AFL—atrial flutter; PAF—paroxysmal atrial fibrillation; PeAF—persistent atrial fibrillation; LsPeAF—long-standing persistent atrial fibrillation; BMI—body mass index; BSA—body surface area; LAD—left atrial diameter; LVMI—left ventricular mass index; LV EF—left ventricular ejection fraction.

**Table 2 medicina-57-00618-t002:** Electrical characteristics of the study population according to arrhythmia type.

	AT *n* = 127	AFL *n* = 61	PAF *n* = 116	PeAF *n* = 260	LsPeAF *n* = 119	*p*-Value
Successful case (%)	121 (95)	58 (95)	112 (97)	242 (93)	104 (87)	0.046
Initial energy (J)	30–70	50–70	50–100	70–100	70–100	
Mean initial energy (J)	57 ± 11	63 ± 13	85 ± 18	90 ± 16	97 ± 12	<0.001
Initial impedance (Ω)	66 ± 13	68 ± 13	68 ± 12	65 ± 11	64 ± 14	0.018
Success energy (J)	66 ± 28	75 ± 29	97 ± 35	115 ± 39	124 ± 39	<0.001
Success current (mA)	16 ± 4	17 ± 5	20 ± 5	23 ± 6	25 ± 7	<0.001
Success impedance (Ω)	70 ± 28	68 ± 12	65 ± 16	64 ± 13	63 ± 13	<0.001
Prior AAD (%)	12 (9.4)	3 (4.9)	4 (3.4)	27 (10.4)	23 (19.3)	0.001

AT—atrial tachycardia; AFL—atrial flutter; PAF—paroxysmal atrial fibrillation; PeAF—persistent atrial fibrillation; LsPeAF—long-standing persistent atrial fibrillation; AAD—antiarrhythmic drug.

**Table 3 medicina-57-00618-t003:** Baseline and electrical characteristics of the patients with high (>70 Ω) and low (<70 Ω) impedance.

Success Group	Impedance < 70 Ω *n* = 421	Impedance > 70 Ω *n* = 216	*p*-Value
Age (years)	59 ± 11	59 ± 11	0.733
Sex, male (%)	383 (91)	137 (63)	<0.001
BMI (kg/m^2^)	25 ± 3	27 ± 4	<0.001
BSA (m^2^)	1.8 ± 0.1	1.8 ± 0.1	0.971
Hypertension (%)	137 (30)	82 (37)	0.036
Diabetes mellitus (%)	33 (7)	23 (10)	0.102
Thyroid disease (%)	36 (8)	24 (11)	0.125
Chronic lung disease (%)	7 (2)	4 (2)	0.504
Valvular disease (%)	23 (5)	13 (6)	0.379
Chronic kidney disease (%)	10 (2)	4 (2)	0.502
Cerebrovascular accident (%)	24 (5)	11 (5)	0.526
Coronary artery disease (%)	12 (3)	14 (6)	0.030
Additional AAD to restore SR (%)	56 (12)	13 (6)	0.010
LAD (mm)	44 ± 6	46 ± 12	0.023
LVMI (g/m^2^)	97 ± 25	93 ± 25	0.063
LV EF (%)	52 ± 9	53 ± 8	0.245
Electrical current (mA)	23 ± 18	18 ± 5	<0.001
Total energy for restoring SR (J)	165 ± 174	186 ± 216	0.187
Number of shocks (*n*)	1.6 ± 1	1.8 ± 1	0.085

BMI—body mass index; BSA—body surface area; AAD—antiarrhythmic drug; LAD—left atrial diameter; LVMI—left ventricular mass index; LV EF—left ventricular ejection fraction; SR—sinus rhythm.

## Data Availability

The data presented in this study are available on request from the corresponding author.

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
