# Peer review of "The Impact of Personal Thoracic Impedance on Electrical Cardioversion in Patients with Atrial Arrhythmias"

_medicina, 2021, doi:10.3390/medicina57060618_

Round 1
Reviewer 1 Report
In this manuscript the authors aim to assess the clinical implications of thoracic impedance in DCCV for various atrial arrhythmias, analyzing a total of 683 DCCVs were performed on 466 atrial tachyarrhythmia patients. They conclude that thoracic impedance was one of the factors that impacted the level of electrical current required for successful DCCV in patients with atrial arrhythmias.
The findings described have been described previously. Moreover, the finding that thoracic impedance was one of the factors that impacted the level of electrical current required for successful DCCV is obvious because, using biphasic selected energy, a constant-current first phase is produced by automatically adjusting the internal resistance of the defibrillator circuit on the basis of the patient’s transthoracic impedance, which is automatically determined at the onset of shock delivery. Moreover, an impact of the type of atrial arrhythmias on thoracic impedance would only be expected if it affected variables related to impedance, such as BMI or overload volume.
On the other hand, the authors do not justify their findings in the Discussion. They should properly discuss what they attribute their results to. The authors should assess what their work contributes to the literature and focus on adequately discussing these aspects.
Other comments:
The authors describe that they assessed the effect of body surface area, but the results are not described in the text nor the tables.
Some comments about the possible complications related to delivered energy are too exaggerated (Introduction, lines 55-57; Discussion, lines 228-230). The references cited in these comments correspond to very old publications.
Author Response
In this manuscript the authors aim to assess the clinical implications of thoracic impedance in DCCV for various atrial arrhythmias, analyzing a total of 683 DCCVs were performed on 466 atrial tachyarrhythmia patients. They conclude that thoracic impedance was one of the factors that impacted the level of electrical current required for successful DCCV in patients with atrial arrhythmias.
The findings described have been described previously. Moreover, the finding that thoracic impedance was one of the factors that impacted the level of electrical current required for successful DCCV is obvious because, using biphasic selected energy, a constant-current first phase is produced by automatically adjusting the internal resistance of the defibrillator circuit on the basis of the patient’s transthoracic impedance, which is automatically determined at the onset of shock delivery. Moreover, an impact of the type of atrial arrhythmias on thoracic impedance would only be expected if it affected variables related to impedance, such as BMI or overload volume.
On the other hand, the authors do not justify their findings in the Discussion. They should properly discuss what they attribute their results to. The authors should assess what their work contributes to the literature and focus on adequately discussing these aspects.
We appreciate the many comments and suggestions provided by reviewer #1. In addressing them, the manuscript has been significantly improved.
That we try to present in our clinical trial is that repeated shocks can cause complications or subclinical damage. Thoracic impedance is associated with successful threshold of initial energy. Female or patients with high BMI have an increased thoracic impedance, which requires consideration when setting up an initial energy. We added the sentence in discussion section to focus our result.
Line 240-244
4.1. The clinical implication of thoracic impedance for successful cardioversion
If the DCCV with same energy is used repeatedly, the current can be increased as a decrease of impedance, which can make the cardioversion successful. To restore sinus rhythm has advantages in cardiac physiology and relieves symptoms, but may also result in shock-induced inflammation and cardiac stunning. This may lead to a temporary deterioration of heart failure even in sinus rhythm status after DCCV.
Line 254-255
4.2. The predictor of thoracic impedance
In previous studies, multivariate analysis demonstrated that large BMI and female gender are independent predictors of high thoracic impedance.
- In this study, multivariate analysis demonstrated that large BMI and female gender are independent predictors of high thoracic impedance.
Line 271-276
4.3. The appropriate level of electrical current for cardioversion
If cardioversion was performed to female or the patient with large BMI patients, using higher initial energy is considered. To reduce the damage of myocardium or skin, the number of DCCVs must be reduced. Sometimes method that increases the contact of the heart with the pad or patch should be used. Furthermore, the ability to measure the thoracic impedance between patches in AED machines will help determine the initial energy.
Other comments:
The authors describe that they assessed the effect of body surface area, but the results are not described in the text nor the tables.
Thank you for this important comment. We added body surface area to Table 1 and Table 3.
Some comments about the possible complications related to delivered energy are too exaggerated (Introduction, lines 55-57; Discussion, lines 228-230). The references cited in these comments correspond to very old publications.
Thank you for this valuable comment. DCCVs by experienced clinicians do not often induce complications. However, complication can increase by repeated shock or excessive energy in inexperienced lab. In a paper recently published, 887 patients with atrial fibrillation who underwent cardioversion in emergency room were analyzed. Among them, 6%, 10%, 1%, 0.1% patients suffered electrical complication, shock or desaturation, chest wall burn and stroke as a complication of DCCVs, respectively. We added sentence presented recent evidence and reference to introduction section.
Line 56-62
DCCVs performed by experienced clinicians rarely induce complications. However, complications can increase by repeated shock or using excessive energy in inexperienced lab. In a recently published paper, 887 patients who underwent cardioversion for AF in emergency room were analyzed. In overall population, 6%, 10%, 1% patients suffered electrical complication like arrhythmia, shock or desaturation, chest wall burn as a complication of DCCVs, respectively.
Added Reference 11: Am J Emerg Med. 2021 Apr;42:115-120.
Line 229-231
Longer periods of sedation may also cause malfunctioning of implanted devices and dermal injuries
- They may also cause malfunctioning of implanted devices in patient with intracardiac devices and dermal injuries
Reviewer 2 Report
In this article entitled “The Impact of Personal Thoracic Impedance on Electrical Cardioversion in Patients with Atrial Arrhythmias” by Seung-Young Roh et al., the investigators assessed the clinical implications of thoracic impedance in DCCV for various atrial arrhythmias. The topic is interesting, but not innovative in my opinion. I appreciate the systematic mode in which the discussion section is performed, it makes the paper easier to perceive. There is soundness to the author's idea, but I have some concerns.
Major revision:
- The evidence in lines 67-68 is no longer relevant. Thoracic impedance varies with many factors, including body weight, body surface area, paddle/pad size, pad location, pad–skin contact, body fat, number of prior shocks, and hypothermia. I recommend you to discuss this in the introduction section so that the results of your study should not appear revolutionary.
- A flow chart will improve following the enrollment process of patients in the study. It will help know, for example, how many participants were excluded after an atrial thrombus was diagnosed by TEE.
- The method used for the measurement of thoracic impedance is not described. It was measured automatically by the defibrillator pads?
- Based on the median BMI, the results of the study cannot be generalized for obese patients. This needs to be added to the limitations section.
- Lines 124-126: The number of patients with hypertension included in the study group presented in the text is different from those described in Table 1.
- Lines 199-202: The average number of patients characterized by BMI, LA diameter, hypertension, and coronary artery disease along the two study subgroups presented in the text are discording from the data presented in Table 3.
- Lines 148, 155-157, and 215-217 are table footnotes and they must be included in the table (they should not be encountered as lines).
- Please clarify one aspect from the heading of Table 3. What does the number included in parentheses after the number of patients from the two subgroups (91, respectively 97) represent?
Minor revision:
- Lines 58, 71, 91, 108, 118, 218, 231, 241, 262, 290: I do not consider necessary the additional spaces that are added before the sections and between a few paragraphs.
- Please revise the way through which bibliographic references are added to the text. The point sign must be after the right brackets.
- Line 79: you mentioned five groups, but you named only four.
- Line 102: The LsPeAF abbreviation is not explained above.
- Line 122: The median age for total patients is discording in the text compared to the data from Table 1.
- Line 123: The authors said there was no difference between the five groups in the body surface area (BSA), but this characteristic did not appear in the table.
- Lines 182, 188: Kindly recommend replacing `AP diameter of LA` and `AP diameter` with LAD for the homogeneity of the paper.
- Line 221: The AAD abbreviation must be explained at the first appearance in the text, even if it is explained in the footnote of Tables 2 and 3.
- Lines 260-261: The meaning of this text is redundant. I don’t consider it helpful, it was already said in the previous sentence.
Author Response
In this article entitled “The Impact of Personal Thoracic Impedance on Electrical Cardioversion in Patients with Atrial Arrhythmias” by Seung-Young Roh et al., the investigators assessed the clinical implications of thoracic impedance in DCCV for various atrial arrhythmias. The topic is interesting, but not innovative in my opinion. I appreciate the systematic mode in which the discussion section is performed, it makes the paper easier to perceive. There is soundness to the author's idea, but I have some concerns.
We deeply appreciate the many comments and the review provided by reviewer #2. These as well, significantly improved the clarity and content of the manuscript.
Major revision:
The evidence in lines 67-68 is no longer relevant. Thoracic impedance varies with many factors, including body weight, body surface area, paddle/pad size, pad location, pad–skin contact, body fat, number of prior shocks, and hypothermia. I recommend you to discuss this in the introduction section so that the results of your study should not appear revolutionary.
Thank you for this important comment. The paragraph was fixed as you stated and we added contents in introduction section.
Line 70-73
Deleted sentence
However, the current-based concept is not applied in practice because factors impacting thoracic impedance are unknown.
Thoracic impedance is known to depend on several factors, including body surface area, pad size, pad location, pad to skin contact. It means that there is a difference depending on the component and the physical distance between both pads.
A flow chart will improve following the enrollment process of patients in the study. It will help know, for example, how many participants were excluded after an atrial thrombus was diagnosed by TEE.
Thank you again. We added flowchart to Figure 1. Only one patient was excluded due to left atrial appendage thrombus revealed by TEE.
Figure 1. Flowchart of study
- The method used for the measurement of thoracic impedance is not described. It was measured automatically by the defibrillator pads?
Thank you for this important comment. It was automatically measured. Since it is not possible to measure before delivering electrical energy. It was only known after shock. We added this content to Method section.
Line 107-108
Thoracic impedance was automatically measured between both patches after delivering energy. There is no additional work to measure thoracic impedance.
Based on the median BMI, the results of the study cannot be generalized for obese patients. This needs to be added to the limitations section.
Thank you for this valuable comment. As you mentioned, obese patient was not analyzed separately but only directly proportional tendency was observed. We added that sentence to limitation.
Line 289-291
Study Limitation
The BMI used in this study are the median value. These results cannot be generalized because obese patients (BMI≥30) was not analyzed separately.
Lines 124-126: The number of patients with hypertension included in the study group presented in the text is different from those described in Table 1.
Thank you again. We fixed the sentence in Result section.
Line 131-132
The number of patients with hypertension was lower in AF groups (PAF, PeAF, LsPeAF) than in AT and AFL groups (p=0.005).
- The number of patients with hypertension was difference in each group.
Lines 199-202: The average number of patients characterized by BMI, LA diameter, hypertension, and coronary artery disease along the two study subgroups presented in the text are discording from the data presented in Table 3.
Thank you for noticing the error. We revised the expression as you suggested
Line 201-205
The high impedance group showed a higher average BMI (27.2±3.6 vs. 24.7±2.8, p<0.001) and a larger average LA diameter (46.1±11.7 vs. 44.4±6.2) than the low impedance group. The prevalence of hypertension (36.9% vs. 29.7%, p=0.036) and coronary artery disease (6.3 vs. 2.6, p=0.01) was also higher in the high impedance group compared to the low impedance group.
- The high impedance group showed a higher average BMI (27±4 vs. 25±3, p<0.001) and a larger average LA diameter (46±12 vs. 44±6) than the low impedance group. The prevalence of hypertension (37% vs. 30%, p=0.036) and coronary artery disease (6% vs. 3%, p=0.01) was also higher in the high impedance group compared to the low impedance group.
Lines 148, 155-157, and 215-217 are table footnotes and they must be included in the table (they should not be encountered as lines).
We revised the sentence as you suggested.
Please clarify one aspect from the heading of Table 3. What does the number included in parentheses after the number of patients from the two subgroups (91, respectively 97) represent?
Thank you for noticing the error. The numbers in parentheses in Table 3 are percentage of the total group. The numbers 91 and 97 at the top of the table were deleted as they cause confusion.
Minor revision:
Lines 58, 71, 91, 108, 118, 218, 231, 241, 262, 290: I do not consider necessary the additional spaces that are added before the sections and between a few paragraphs.
Thank you again. As your recommendation, we fixed it.
Please revise the way through which bibliographic references are added to the text. The point sign must be after the right brackets.
Thank you for this comment. We revised the bibliographic references in text.
Line 79: you mentioned five groups, but you named only four.
Thank you again. We added one, long-standing persistent atrial fibrillation group.
Line 84-87
Atrial tachycardia (AT), atrial flutter (AFL), paroxysmal atrial fibrillation (PAF, AF that spontaneously terminated), and persistent atrial fibrillation (PeAF, AF persisting beyond 7 days).
- Atrial tachycardia (AT), atrial flutter (AFL), paroxysmal atrial fibrillation (PAF, AF that spontaneously terminated), persistent atrial fibrillation (PeAF, AF persisting beyond 7 days) and Long-standing persistent atrial fibrillation (LsPeAF, AF persisting beyond 1 year)
Line 102: The LsPeAF abbreviation is not explained above.
Thank you for this comment. We added the explanation of LsPeAF abbreviation.
Line 84-87
Atrial tachycardia (AT), atrial flutter (AFL), paroxysmal atrial fibrillation (PAF, AF that spontaneously terminated), and persistent atrial fibrillation (PeAF, AF persisting beyond 7 days).
- Atrial tachycardia (AT), atrial flutter (AFL), paroxysmal atrial fibrillation (PAF, AF that spontaneously terminated), persistent atrial fibrillation (PeAF, AF persisting beyond 7 days) and Long-standing persistent atrial fibrillation (LsPeAF, AF persisting beyond 1 year)
Line 122: The median age for total patients is discording in the text compared to the data from Table 1.
Thank you again. As your recommendation, we fixed errors.
Line 128-129
We performed 683 total DCCVs in 466 enrolled patients with atrial arrhythmias (59.5±11.3 years old, 376 males).
- We performed 683 total DCCVs in 466 enrolled patients with atrial arrhythmias (60±11 years old, 376 males).
Line 123: The authors said there was no difference between the five groups in the body surface area (BSA), but this characteristic did not appear in the table.
Thank you for this comment. We added body surface area value in Table 1 and 3.
Lines 182, 188: Kindly recommend replacing `AP diameter of LA` and `AP diameter` with LAD for the homogeneity of the paper.
Thank you again. The expression of ‘Left atrial anteroposterior diameters’ and ‘AP diameter of the LA’ were fixed to ‘LAD’
Line 221: The AAD abbreviation must be explained at the first appearance in the text, even if it is explained in the footnote of Tables 2 and 3.
Thank you for this comment. We added explanation of ‘AAD’ abbreviation to Discussion
Lines 260-261: The meaning of this text is redundant. I don’t consider it helpful, it was already said in the previous sentence.
Thank you again. As your recommendation, we finally deleted this sentence.
Deleted sentence
Apart from BMI, fatty tissue of the breasts may also act to increase thoracic impedance in woman.

Round 2
Reviewer 1 Report
The authors state that "the BMI used in this study are 289 the median value. These results cannot be generalized because obese patients (BMI≥30) was not analyzed separately." ("were")
Why not? This analysis can be performed easily.
Reviewer 2 Report
No further comments. The authors have addressed all my comments and improved the manuscript accordingly.